# Edema after CNS Trauma: A Focus on Spinal Cord Injury

**DOI:** 10.3390/ijms24087159

**Published:** 2023-04-12

**Authors:** Mostafa Seblani, Patrick Decherchi, Jean-Michel Brezun

**Affiliations:** Aix Marseille Univ, CNRS, ISM, UMR 7287, Institut des Sciences du Mouvement: Etienne-Jules MAREY, Equipe «Plasticité des Systèmes Nerveux et Musculaire» (PSNM), Parc Scientifique et Technologique de Luminy, CC910-163, Avenue de Luminy, F-13288 Marseille, CEDEX 09, France

**Keywords:** edemas, traumatic spinal cord injury, central nervous system

## Abstract

Edema after spinal cord injury (SCI) is one of the first observations after the primary injury and lasts for few days after trauma. It has serious consequences on the affected tissue and can aggravate the initial devastating condition. To date, the mechanisms of the water content increase after SCI are not fully understood. Edema formation results in a combination of interdependent factors related to mechanical damage after the initial trauma progressing, along with the subacute and acute phases of the secondary lesion. These factors include mechanical disruption and subsequent inflammatory permeabilization of the blood spinal cord barrier, increase in the capillary permeability, deregulation in the hydrostatic pressure, electrolyte-imbalanced membranes and water uptake in the cells. Previous research has attempted to characterize edema formation by focusing mainly on brain swelling. The purpose of this review is to summarize the current understanding of the differences in edema formation in the spinal cord and brain, and to highlight the importance of elucidating the specific mechanisms of edema formation after SCI. Additionally, it outlines findings on the spatiotemporal evolution of edema after spinal cord lesion and provides a general overview of prospective treatment strategies by focusing on insights to prevent edema formation after SCI.

## 1. Introduction

What does edema mean? Here is a look into the term’s extensive use:

Edema is one of the oldest medical terms still in use today. This term encompasses various pathophysiological observations with the only common factor being an increase in water content. In the Corpus Hippocraticum, the term edema was used to describe all types of swelling and was considered to result from a lack of liquid elimination [1]. Along with redness, warmth, swelling and pain, edema has been reckoned, since Celsius, as the main sign of inflammation [2]. Throughout history, understanding the pathophysiology of edema evolved along with the theory of secretion and absorption, cell theory, and physiology advances [3]. Underlying mechanisms largely differ depending on organs, tissues, local microcirculation, and mechanisms of water influx and accumulation.

Nevertheless, this term is still applied to describe symptoms that exhibit similarities but a great degree of divergence in pathophysiology and clinical considerations such as brain and spinal cord edema, macular edema, Reinke edema, pulmonary edema, muscle edema. Nowadays, edema has not deviated much from its original definition. It is defined as a consequence of imbalance in forces that control fluid homeostasis within cells, interstitial spaces, and capillaries [4]. Even though it is about one single term in all tissues, the pathophysiology of edema leads us to consider various particularities of different types and subtypes of edema.

Brain edema was described by Robert Whytt as an accumulation of water due to abnormal vasculature [5]. Later, Martin Reichhardt suggested different terms to describe an accumulation of water that increases brain volume outside cells and merely inside cells [6,7]. This slight difference, along with the heritage of the traditional Monro–Kellie doctrine, inspired Igor Klatzo to frame his famous classic distinction between vasogenic and cytotoxic edema in the brain [8], which will be discussed later. Since then, cerebral edemas (CEs) after traumatic brain injury (TBI) have been the main model for studying the genesis and evolution of edemas in the CNS.

CEs are complex and heterogeneous processes that result from traumatic or non-traumatic events and represents the most common cause of increased intracranial pressure (ICP), along with other factors such as expanding hematoma. As a consequence, it is necessary to distinguish between edema as an increase in tissue volume due to an increase in water content, and swelling—a change in volume tissue caused by different factors, which may or may not include edema [9].

Brain edema can be life-threatening, as it potentially causes damage to the brain tissue due to interruption of the blood flow after an increased intracranial pressure (ICP). Traumatic CE depends on the nature of the initial lesion, the affected area, and the additional secondary mechanisms. Whether in the brain [10,11] or in the spinal cord [12], edemas lead to necrosis by direct compression and by its resultant hemorrhages, hypertension and ischemia. All of these impair the local functions of nervous, glial, and vascular tissues. The disruption of extracellular homeostasis, as well as the alteration of astrocytic functions due to microenvironment imbalance and water accumulation, worsen excitotoxicity [13,14]. It is also associated with glial activation which can exacerbate the inflammatory cascade and enhance the secondary injury. Consequently, clinical findings correlate edemas with a mortality increase up to 60% after brain injury [15]. Furthermore, spinal cord edema is correlated with a severe impairment scale and poor recovery at follow-up [16,17,18].

Despite decades of research, mechanisms underlying brain edemas are not fully elucidated, and no targeted therapy is clinically available. However, the brain remains better investigated and understood as the model of edema formation in the CNS (for a recent review [15,19], and for a recent book [20]) in comparison to the spinal cord, with the latter being relatively less known.

## 2. Edema in the CNS

### 2.1. Edema Classification in the CNS

The classification of edemas in the CNS was first characterized in the brain and then extrapolated to the spinal cord. It has evolved alongside progress in understanding the underlying mechanisms of edemas. For instance, the Reichhardt distinction between brain swelling (Hirnschwellung) and brain edema (Hirnödem) was based on a morphological description of the nervous tissue being dry and tough for the former and molt and soft for the latter [6]. Later, the Klatzo classification system for brain edemas, which is still widely used, relies on origins, general mechanisms, and localization of the edema. This classification divides the condition into two types: vasogenic edema (also called angiogenic edema) and cytotoxic edema (also called cellular edema). The vasogenic edema is driven by hydrostatic forces when a fluid from a blood vessel enters brain tissue due to a blood brain barrier (BBB) defect, while cytotoxic edema is driven by ionic forces and results after a loss of homeostatic gradients that implicates a water accumulation within the cell body, mainly in astrocytes [8]. Klatzo emphasized that these two classes, although different, are often coexistent and not mutually exclusive [21].

Later, along with the development of molecular biology, a mechanistic approach influenced researchers to expand Klatzo’s categories. Three new classes were added to those of Klatzo’s: hydrostatic edema, interstitial edema, and hypoosmotic edema [22,23,24]. The hydrostatic edema is a protein-free plasma ultrafiltrate resulting from an imbalance between the forces that drive fluid movement across the walls of capillaries. The interstitial edema, previously described by Fishman et al. [25], results from the invasion of cerebrospinal fluid (CSF) through ependymal cells into extracellular space. Lastly, the hypoosmotic edema takes place when the concentration of solutes in the extracellular fluid is lower than normal. However, some researchers, who are still referring to Klatzo’s theory, consider these new labels as pointing out etiology rather than physical location [26]. Other researchers, under the advancement of our understanding of pathophysiology alongside clinical observations, consider Klatzo’s classification to be unable to establish an integral system of description of the complex blended types. Jha et al. even consider that classifying edema as vasogenic or cytotoxic, although still used, is nowadays artificial [15]. After advances in understanding edema mechanisms, new and different modifications of this scheme have been suggested, since it is not only about the coexistence of these two types, but rather about the interdependency of the underlying processes. Rather than considering two categories, a perspective regards edema as a condition that can start inside or outside cells and can spread to other areas. It is caused by nested mechanisms that may be qualified as cellular, vascular, and/or combined in different proportions [27]. In addition, Young and Constantini suggested revising the traditional dual classification in the light of understanding new molecular mechanisms resuming Miller’s distinction [28]. Thus, they added a third category designated to ionic edema, which refers to the process of water uptake from capillaries through trans-endothelial routes with an intact blood–brain barrier. Ionic edema occurs when there is an imbalance in the concentration of fluids between the blood vessels and the surrounding tissues in the brain. This type of edema shares differences and similarities with both cytotoxic and vasogenic edema and represents a continuum of transition between the two of them [29]. Since then, this category has been adopted by different researchers for brain edema [30,31,32] and spinal cord edema [33,34].

The progression of these three types of edemas does not follow a specific temporal sequence, but occurs concurrently. Differential progression of these types can be distinctly monitored by imagery techniques to determine the spatiotemporal profile of each category (for recent review [35]). This may be important to further understand eventual temporal, sequential, and mechanistic nuances between brain and spinal cord in formation and clearance of edemas, given the slight dissimilarities between them. This may help in developing more effective and refined protocols in preclinical studies, as well as better classification of patients for new therapeutic tests in clinical trials. Different suggested classifications are summarized in Table 1.

### 2.2. Local Particularities of Edema in the Spinal Card

Edema formation is one of the first events of the acute phase of spinal cord injury (SCI). As for CE, spinal cord edema may differ depending on the type of the initial injury (laceration, ischemia, dislocation, compression, contusion, torsion…) and its subsequent postlesional inflammatory events. While the mechanisms of edema formation in the brain and the spinal cord are quite similar, there are some structural and tissue-specific differences that can affect edema formation in these two regions. These differences are depicted in Figure 1.

The brain and spinal cord are both contained in a rigid outer shell, where volume and pressure control are important to avoid compartment-like syndrome. Edema increases intraspinal pressure (ISP) after SCI [12,36] and intracranial pressure (ICP) after TBI [37]. According to an advanced understanding of the traditional Monro–Kellie hypothesis, this leads to changes in the parenchymal volume to maintain the same dynamic equilibrium within the rigid skull. Being large, wrinkled, and folded in depth within the skull, the brain tissue compliance for freed-up strategies by herniation may be different from those of a thin cylindrical structure as the spinal cord. It is well known that the volume-pressure curve is non-linear in the brain [19] and in the spinal cord [38]. In both cases, the trend of the volume–pressure curve stays relatively constant up to a specific point, after which it increases exponentially. However, the differences in the space between parenchyma and the skull may influence the time point at which the pressure, and its subsequent evolution, sharply rises. For instance, the vertebral canal gives longitudinal space for the expansion of swollen tissue in the spinal cord [39,40] that also spreads away from the site [41].

Moreover, in the brain, the central ventricles represent large, interconnected cavities. However, in the spinal cord the ependymal canal is thinner, and it generally becomes occluded over age, which highlights an important anatomical difference.

Compared to the brain, the subarachnoid compartment around the spinal cord is relatively large in comparison to the organ, which gives enough space for circumferential tissue expansion when it occurs only in the site of injury [42]. Furthermore, unlike the cranium, a physiological epidural space exists between the osteofibrous boundary of the dura mater of the spinal cord and the spinal cord [43]. All these factors may influence the evolution of post-trauma tissue pressure and distinguish the non-linearity in the brain and in the spinal cord. A more thorough kinetics comparison is still required.

On the histological level, there are some differences between the structure of the nervous parenchyma and the meninges in both the brain and the spinal cord that can affect edema formation. For instance, given that white matter tracts predominate in the spinal cord, it has been suggested that, unlike the brain, the evacuation of edema in the spinal cord can occur through intracellular or extracellular routes along the white matter tracts [42]. In addition, on the meningeal level, the arrangement of collagen and elastin fibers are more oriented in the spinal cord’s dura matter than in the brain’s, which suggests that the latter may be able to relax faster than the former [44,45]. Another relevant consideration is that the spinal pia mater is thicker, more compact, and less vascular than the cranial pia mater, and is often referred to as a fibrous layer [46]. The role of spinal meninges after SCI is remarkably underestimated compared to cranial meninges after TBI, and it has been suggested that they have different tissular and biomechanical properties that have been reviewed [43].

Edema is normally evacuated into the subarachnoid and ventricular CSF and into the blood. Anatomical and functional differences in these interfaces in the brain and the spinal cord may affect the evacuation of water excess. The blood–spinal cord barrier (BSCB) represents more than the analogous and morphological extension of the blood–brain barrier (BBB). Despite similar cellular and molecular structures, there are slight but important structural differences between these two barriers that could be responsible for dissimilarity in the 5regulatory role, in degree of permeabilization and in resulted water influx after damage (for recent review [47]). Perivascular pericytes, that are crucial for maintenance of endothelial barrier properties, are lower in the BSCB than in BBB [48]. In addition, the large superficial vessels around the spinal cord are richer in glycogen, which affects microhemorrhages after the initial impact as well as the supply of energy storage. Additionally, BSCB is less tightly packed and more permeable than BBB [49] due to differences in tight junction proteins, such as zonula occludens-1 and occludin, and adherence junctions such as cadherin and b-catenin [50]. This variance in the proteomic expression can affect postlesional immune cell infiltration, spread of pro-inflammatory molecules, transendothelial transfer, and the impact and evolution of subsequent edema formation. It has been reported that in the spinal cord, immune cell infiltration after injury is instinctively more important than in the brain. This makes the inflammatory response after spinal injury broader, and the spinal cord more sensitive to secondary events than the brain, especially in the first 24 h after the injury [51,52,53].

In addition to the BBB and BSCB, the plexus choroid represents a second interface between the CNS and fluid circulation. Recent work revealed that despite homogeneity, ependymal cells in the brain and spinal cord can be distinguished by gene signatures and can play distinct roles in lymphatic clearance and during inflammation [54].

These examples may also suggest the existence of subtle differences in the mechanisms of water accumulation and clearance after injuries that are to be investigated. More research is needed to consider the impact of these physiological, structural, and biomechanical aspects on the mechanisms of edema formation after SCI and TBI.

## 3. Edema Formation after SCI: A Loop of Interconnected Mechanisms

The mechanisms contributing to edema after SCI or TBI are various and interconnected. An exhaustive survey of all suggested mechanisms is beyond the scope of this paper; however, here, as shown in Figure 2, we focus on examples that reveal the interdependence of vasogenic and cytotoxic mechanisms of edema that may be carried out simultaneously. Early inflammatory cytokines can enhance BSCB mechanical rupture and subsequent vasogenic edema [55,56,57] and can also trigger cellular edema. For example, IL-6 is suggested to lead to the breakdown of the blood–brain barrier [58]. At the same time, this inflammatory molecule can trigger a signaling process involving HMGB1 (High mobility group box 1) protein that leads to increased production of AQP4 (Aquaporin-4) channels in the endfeet of astrocytes, causing a harmful cellular swelling [59]. In addition, other inflammatory molecules can participate differently in edema mechanisms depending on the pathway. For instance, histamine contribute to vasodilation, BSCB integrity loss and vasogenic edema formation [60]. However, the same molecule has been suggested to trigger AQP4 internalization in astrocyte, which can thwart cellular permeability for water [61]. Nonetheless, this mechanism has not been fully investigated in the dynamics of edema after SCI.

Furthermore, the generation of free radicals that can result from cytotoxic edema leads to microvascular injury, endothelial apoptosis, and disruption of endothelial tight junction protein complexes that can exacerbate vasogenic edema. Likewise, oxidative stress after cell swelling also leads to increased expression of the degrading enzymes MMPs (Matrix metalloproteinases) in perivascular cells and infiltrating or resident inflammatory cells, leading to further degradation of endothelial tight junctions, basement membranes, and cell adhesion molecules that can, in turn, lead to vasogenic edema. Furthermore, vasogenic edema resulting from early damage of the BSCB can be pursued by ionic flux due to changes in channel expression, such as NKCC1 (Sodium potassium chloride cotransporter 1), even after BSCB is restored [62]. In addition, disruption of BSCB leads to impaired oxygen and glucose supply, and ATP depletion that causes active pump failure, passive transport dysfunction, and opening of other channels such as the nonselective cation channel NC_Ca-ATP_, thus eliciting cytotoxic edema [63,64,65]. These examples illustrate the reciprocal causality in cytotoxic and vasogenic mechanisms.

AQPs represent another telling and challenging example to show this interrelatedness. In the CNS, these bidirectional water channels are located mainly in astrocyte foot processes and determine the rate at which osmotic water moves across the BBB/BSCB and inside cells (reviewed in [33,66,67]). They regulate cell volume homeostasis, and when dysregulated and overexpressed after SCI, they play a central role in the formation of edema. There is an ongoing debate about the role of AQPs in the development of edema due to conflicting findings about how AQPs contribute and alleviate different forms of edema, which is related to the dual, biphasic, and complex role of these channels. For example, different studies consider the AQPs to play a role in the formation of cytotoxic edema [68], while others consider a facilitating role in the water’s movement from the circulation into spinal tissue, and that the deletion of AQPs can protect the integrity of interfaces between the CNS and the circulation [69]. AQPs are also related to neuroinflammatory events [70] which can exacerbate the BSCB permeabilization. While some authors consider that increasing AQPs in the early stages of SCI help to reabsorb water and reduce vasogenic edema, with the possibility of worsening the formation and progression of cytotoxic edema in later stages [71], others believe that inhibiting AQPs’ function shortly after SCI, but not later, may be beneficial [72]. Furthermore, there is growing evidence on the role of AQPs in eliminating vasogenic edema [73,74,75]. Even though some researchers believe that edema after SCI is eliminated through mechanisms other than AQPs, and that inhibiting AQPs after SCI, unlike in the brain, have no effect on edema removal [42], others discovered that AQPs promote the removal of excess water after SCI and have beneficial protective effects [76]. The understanding of this dual role in both cytotoxic and vasogenic edema mechanisms is not clear yet. Moreover, edema mediated by AQPs can trigger various signaling pathways that can significantly alter astrocytic functions [77] such as glutamate reuptake failure [78,79] which, in turn, can lead to an exacerbated cytotoxic [56,80] and vasogenic edema [81,82,83]. These examples show that the regulation of AQPs is more complex than previously thought, as it depends on spatial and temporal factors and involves a cascade of interrelated events that can spread in chains of causes.

In addition to all these factors, valuable insights of animal models should be carefully considered when trying to understand real clinical scenarios that are much more combined and heterogeneous. Even a seemingly simple experimental model is more heterogeneous than we initially believed. For example, in the SCI weight drop model, the contusion injury can be precisely controlled, but this can be followed by temporary periods of reduced blood flow (ischemia) and subsequent restoration of blood flow (reperfusion), which is still distinct from a full biphasic ischemia-reperfusion injury. This model is the most common in SCI animal studies [84,85]. Despite its simplicity, it is still more complex than other models adopted to study isolated CNS edema categories [86] and represents an acceptable combination of mechanisms normally observed in clinical cases. This makes it relevant to study the different and interconnected mechanisms underlying spinal edema. However, the contusion model requires laminectomy to access to the spinal tissue unlike other models used to study ISP in closed canals [87,88]. This inherent step of the surgery by itself alleviates the impact of edema and the subsequent pressure increase, as reported in clinical findings [89,90,91]. Such an effect can impede the progress of translational research and clinical applications. Nevertheless, it is theoretically possible to improve the contusion model and avoid the biased decompression effect of the laminectomy by synthetically restoring the vertebral column before suturing operated areas.

All of these factors should be considered when attempting to infer the mechanisms, progression, and kinetics of edema development after different types of SCI.

## 4. Time Course of Spinal Cord Edema Formation after Injury

It is well known that there is a proportional relationship between the magnitude of injury and the dynamics of edema formation and spread rostrally and caudally [92]. Edema is progressive by nature, and the rate at which it develops and resolves can vary depending on the nature, severity, and location of the injury. Immediately after SCI, there is often a rapid increase in water content that may last for several hours, reaching a peak during the acute/subacute phase, after which it starts to resolve gradually. The cause of variation of reported time course could be attributed to the species studied, the type of spinal cord injury model employed, and the specific time points chosen in the experiment. Findings using the classic wet/dry weight method to assess water content [93,94], although controversial [95], show relatively consistent results. The literature showed that edema after a thoracic contusion in monkeys starts to increase five minutes after the injury, hitting a relative high point after 6 h, then reaching its maximum intensity in five days, before beginning to subside gradually over the following days [96]. The early increase in water content after the contusion model in rats was closely correlated with the observed increase in spinal cord tissue and the decrease in Mg^2+^ and K^+^ throughout the development of edema, normally associated with cytotoxic edema, with a return to baseline 7 days post injury [97,98,99]. This can be related to cell membrane damage, but also to an upregulation of NKCC1 expression after the injury [62,100,101]. Another study on compression model suggested that leakage and permeability changes, normally associated with vasogenic edema, occurred early after injury [102]. Recently, a paper showed that the water content in the epicenter of the lesion increases immediately one hour after the injury, reaching a peak 72 h after the injury, then decreasing but remaining significantly higher compared to baseline. However, in the rostral and caudal areas adjacent to the epicenter, the water content reaches a significant peak at 72 h and returns to the baseline 7 days after the injury [103], which indicates a heterogeneous spread of edema. It should be noted that in this study, as in other studies [67,71], the edema develops in one wave without a relative peak at 5 h after injury. However, it has been suggested, in a biphasic evolution of water content after a compression model in rabbits, that hemorrhage peaks within 5 h after injury, while edema reaches a maximum 3–7 days later [12]. These findings, being consistent with the time course as suggested by Yashon et al. [96] seem to also be coherent with the results of MRI experiments. In a rat contusion model, a histopathology study of acute SCI with MRI imaging showed that early hemorrhage diffuses throughout the cord as time progressed to reach a maximum at 12 h, then signals correlated with edema formation in vacuolated white matter increases after 48 to 72 h and continued until 96 h [104]. A study in the hyperacute phase reported that the hemorrhage started a few minutes after impact and increased by approximately 0.15% per minute to reach approximately 45% in 5 h, while the edema was observed as early as 12 min after the injury without knowing if it is cytotoxic or vasogenic [105]. Leypold et al. suggested that the hemorrhage increases rapidly early after the injury but tends to level off quickly and be comparatively static. Edema, on the other hand, starts out patchy, eventually becoming more widespread and lasting longer as the secondary injury cascade progresses [41]. Similarly, using a telemetry system to monitor intramedullary pressure in rabbits after compression injury, another study [106] has shown that pressure displays a dynamic change in three phases. This paper showed a sharp increase within the first 7 h after injury, which is mainly related to spinal cord hemorrhage, while later stages are associated with edema formation and permeabilization of BSCB [106]. Even though it is difficult to determine a unique time window for different models and species using different techniques and statistical analyses, various methods have shown that the change in water content after spinal cord injury is dynamic with two distinct phases, the first being in the early hours related to hemorrhage, and the second to edema formation via a combination of vasogenic and cytotoxic mechanisms.

## 5. Toward a Further Treatment of Spinal Cord Edema

For a survey of experimental treatments for edema after SCI, the reader is referred to a recently published systematic review [85]. As stated by Kahle et al., treatments currently in use are still empiric, suboptimal and limited to a management approach [107]. Although there has been extensive research on the causes of cerebral edema, there has been little progress in the development of nonsurgical treatments for brain swelling over the past 50 years. According to medical guidelines for acute SCI management [108,109], current clinical strategies to treat edema in the acute SCI phase are still limited to early surgical decompression and CSF drainage. As recently indicated by the American College of Surgeons Trauma Quality Programs (ACS TQP) Guideline for spine injury 2022, there is no proven effective medication that can successfully treat spinal cord injury edema [110]. Additionally, despite recent efforts [111], there is currently no valid and reliable method for measuring the degree of swelling, which could be used in subsequent studies to evaluate the effectiveness of treatments for reducing edema. Clinical decisions about treatment effects are often inferred by measuring the ICP or ISP. National and international organizations have different recommendations for the use of methylprednisolone, and the potential complications of the medication should be weighed against the potential benefits. Other potential medications have been suggested to mediate edema reduction and have been tested or are currently under investigation in clinical stages for acute management of spinal cord in phases II–III [112,113]. For example, clinical trials have been conducted on GM1 (monosialotetrahexosyl) ganglioside [114], minocycline [115], riluzole [116,117,118] and granulocyte colony stimulating factor [119,120]. However, these are investigated for their neuroprotective and anti-inflammatory effects in acute phase without being linked specifically to edema mechanisms. To date, there is no drug to reduce edema after spinal cord injury, and the recent most-promising molecules are still in the preclinical phase.

Some ongoing studies in animal models are interested in targeting the regulation of edema-specific structures such as AQPs or NKCC1 with new strategies [67,68,72,121]. Others continue to focus on edema decrease related to the general neuroprotective effect of some molecules such as bradykinin antagonist [122], melatonin [123], and a combination of neurotrophic factors [124], as well as improving the classic osmotherapy using an implantable osmotic transport device [103]. Studies which targeted specific edema-related structures have shown promising results in understanding the mechanistic level of edema and have demonstrated edema reduction. AQP4 inhibition, by subcellular delocalization using trifluoperazine, an FDA approved drug [72], remains the most successful strategy to treat edema after SCI in the past 10 years, according to a recent systematic review [85]. However, this strategy and others selectively or partially targeting AQP4 water channels, such as TGN-020 [125], HYP9 [126], and bumetanide [127], show interesting findings that are still limited to preclinical models.

The controversial case of Methylprednisolone highlights stagnation in clinical results and recommendations [112]. To understand why only a limited number succeed in becoming a tested drug, it is important to mention the limitations of models in animal experimentations regarding the nested mechanisms of edema formation, as well as the limitations of techniques to assess edema reduction [95], which rarely include multimodal monitoring. In addition, there is an essential need for a synergetic approach to effectively reduce edema. With the exception of a few studies [68,128], most current studies only focus on one molecule and one category of edemas.

## 6. Conclusions

Despite the importance of preclinical study results, it is worth noting the lack of a synergistic approach that targets different pathways of edema formation. Given the multifactorial and complex characteristics of edema formation, a new attitude is needed to develop efficient pharmacological strategies. Interpreting experimental results based on Klatzo’s traditional classification may limit our understanding of the overlapping mechanisms behind edema. This will continue to hinder the development of synergic strategies to target different pathways of edema formation rather than targeting one of the predominant edema subtypes. Therefore, it may be more useful to classify mechanisms rather than the edema itself, and to consider Klatzo’s subtypes as operational rather than systematic categories. Otherwise, new observations could be theory-laden, which can impede the development of new drugs. Pharmacological strategies to attenuate edema after SCI, taking into consideration spinal cord specificities, are one of the most important steps of acute neurocritical care to preserve spared tissue.

## Figures and Tables

**Figure 1 ijms-24-07159-f001:**
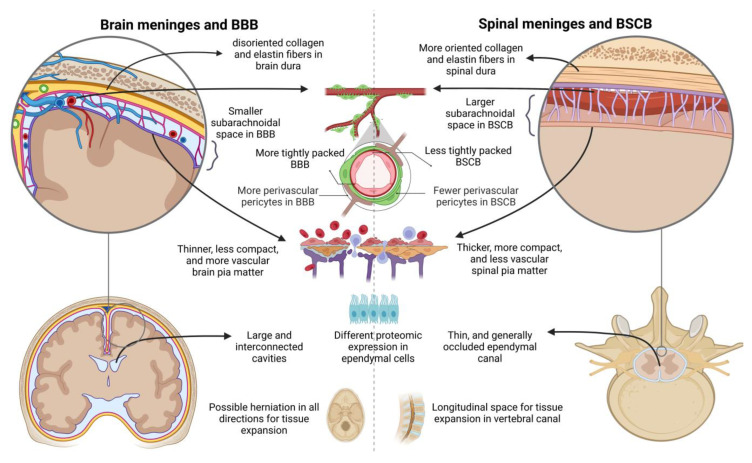
Structural differences in tissue-fluid interfaces and skull shape in brain (left) and spinal cord (right). From bottom to up: freed-up strategies and skull shape, ependymal canal and ventricle cavities, pia matter, subarachnoid space and meninges vascularization, dura matter.

**Figure 2 ijms-24-07159-f002:**
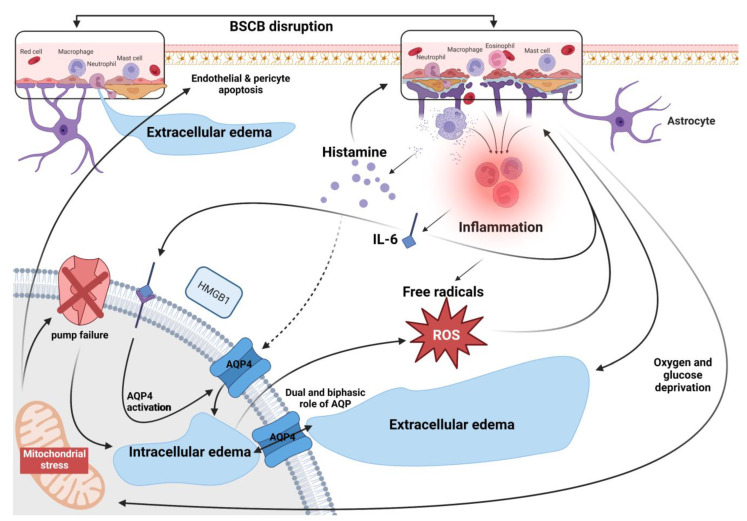
Interconnected mechanisms of vasogenic and cytotoxic processes of edema formation.

**Table 1 ijms-24-07159-t001:** A summary of the main edema classifications and the corresponding categories.

Basis of Classification	Edema Categories	Description
Morpho-histology	Edema	Molt and soft tissue
	Swelling ^1^	Dry and tough tissue
Origins and localizations	Cytotoxic	Intracellular water content
	Vasogenic	Extracellular water content
Etiologies	Cytotoxic	Cell pumps dysfunction
	Vasogenic	Disruption of blood barrier
	Hydrostatic	Vessel wall imbalance
	Interstitial	Ependymal imbalance
	Hypoosmotic	Low plasma osmolality
Mechanisms	Cytotoxic Vasogenic Ionic	Cell pumps disfunction Disruption of blood barrier Through trans-endothelium
Interdependency	Complex conditions	Combination of nested and interdependent mechanisms

^1^ Hirnödem (brain edema) and Hirnschwellung (brain swelling).

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
