# Peer review of "Edema after CNS Trauma: A Focus on Spinal Cord Injury"

_ijms, 2023, doi:10.3390/ijms24087159_

Round 1
Reviewer 1 Report
In this review article, the authors tried to elucidate and summarize the mechanisms of edema after spinal cord injury. Overall, the review article is well-organized and fully explored the mechanisms of the three commonly seen types of edema in spinal cord injury. The illustrations are also very clear and fully explain the complexities of edema in the spinal cord. However, the authors failed to mention the effects of edema in different types of types. Most of the original research cited by the authors uses a contusion-based traumatic injury, which is often performed after laminectomy. How does the laminectomy itself affect the outcome of edema? In clinical settings, the spinal column is usually intact after injury, making increased ICP/ISP and subarachnoid hemorrhaging far more likely to cause detrimental effects than after a laminectomy in experimental settings. Essentially, we had already released some subarachnoid pressure with the laminectomy, did we not? In fact, the authors didn’t delve much into the limitations of preclinical study results in relation to clinical cases of spinal cord neurotrauma. What stage are the more promising treatments in? Why does so few of them make it to fruition as a tested drug? Perhaps the manuscript would be more detailed with some additional discussions on this issue.
Author Response
Thank you for your careful reading and reviewing. We appreciate the time and effort you invested to review and comment our manuscript. In response to your feedback, please find the following:
- Contusion-based model injury is used in the majority of the preclinical spinal cord lesion studies. In fact, it is precisely controlled and it represents the most common pattern of studied injury in animal models (followed by transection, then compression). For this reason, as for clinical reasons, most of the cited research were based on this model. We added a specific paragraph to emphasize this idea (line 269 to 274).
- Indeed, the decompressive laminectomy is clinically recommended after SCI to relieve the intraspinal pressure as you mentioned. Some methods can induce reproducible models of SCI with no need for laminectomy. However, they show less heterogenous mechanisms than contusion models. The optimal choice is to keep the mechanistic heterogenous profile of the contusion model without the biased effect of the decompression laminectomy. Unlike previous studies in which contusion is made after laminectomy, our team generated an accurate model that retores the closed vertebral canal before suturing the operated areas. In order to improve the model, the team designed a sterilized and biocompatible 3D-printed plate to be added onto the exposed spinal cord to cover the injury site after contusion. The plate was piled up with dental cement on both sides of the vertebral bone to restore the vertebral column as in clinical settings. This work has not been published yet and will be submitted soon. Following up on the points you raised we added a paragraph to discuss this aspect (line 274 to 281) .
- To clarify the status of current treatments, and to make a link with the limitations of preclinical models, we added some additional details and discussions to emphasize these points on the light of your comments in order to improve the manuscript (line 352 to 360 and line 368 to 381).
Reviewer 2 Report
This review summarizes the current understanding of the differences in edema formation in the spinal and brain, as well as the outlines findings on the spatiotemporal evolution of edema after spinal cord lesion. This article comprehensively summarizes the central nervous edema which has guiding significance for drug development in the treatment of edema. But there is room for improvement in the content and structure of the article.
1. In part 2.1 of the article, the basis for the classification of edema is not mentioned. This part is more like the opinions of different scientists.
2. The differences and similarities between cerebral and spinal edema can be described separately.
3. The pathophysiology of edema can be described in the introduction section of this article.
4. There are few figures and tables in the review.The pathophysiology, classification, and similarities and differences of edema in the brain and spinal cord can be combined with figures and tables.
Author Response
We would like to thank you for reading our manuscript and for writing this constructive feedback that helped us to strengthen our arguments. To address the concerns you mentioned, please find the following:
- To make it clear, we added additional explanation in order to explicit the principles of classification for each scientist (line 81 to 85 and line 93 to94).
- This paragraph discusses the slight differences which distinguish the local environment of the spinal cord compared to the brain’s one. It is more about the particularities of the spinal cord regarding edema formation, evolution, and resorption than a pure description of differences and similarities. In light of your comment, we decided to change the title of the section to emphasize the main objective of the paragraph (line 133).
- Following up this point, we added a brief part in the introduction to present the general physiopathology of edema in the CNS (line 63 to 71).
- In response to this point, and in complementary with the improvement made in the light of point 1, we added a table to sum up different classifications with a specific column to precise the basis of description for each classification system (line 131 to 132).
Reviewer 3 Report
This is a well-written, tightly focused review on the literature pertaining to the question of the role of edema following spinal cord injury. The authors take the opportunity to compare the evidence for differences between the edema following traumatic brain injury with that following trauma to the spinal cord. The underlying cellular mechanisms that may contribute to such differences are explored and well-illustrated with a single comprehensive figure. The authors also provide a historical context for the terminology on edema that is currently in use, along with valid questions regarding their utility. Although no specific hypothesis is formulated, the review points to various areas that deserve further attention, especially as it pertains to the use of different animal models. In summary, this is a succinct and scholarly review that should be useful for those scientists and clinicians who are seeking to understand the complex changes that occur in the injured spinal cord, especially insofar as they differ from those occurring in the injured brain. There are a few corrections needed to the language (the title, for example, needs to be clarified) but these are relatively minor in scope. (I must confess that I have never had to write a paper in a language foreign to me, so I admire the mostly masterful use of English here.)
Author Response
We would like to thank you for taking the time to read the manuscript and for giving us your insightful comments. We are grateful for your feedback.
In response to the point you raised, we changed the title of the review in order to make it clearer. We corrected some mistakes in order to improve the language.
Round 2
Reviewer 2 Report
The article has been revised as previously suggested. In general, the structure and content of the revised article are satisfactory. But there are still some flaws that we hope the author to improve.
1. The title of the article needs improvement. Most content of the article has little relevance to the title.
2. Part 3 (Edema formation after SCI: a loop of interconnected mechanisms)of the article can be illustrated with a schematic.
Author Response
We are grateful for your feedback. Thank you for your new comments. We have taken your remarks regarding the title and your proposal to add an adapted illustration relative to the third part.
1 – We propose the following title: « Edema after CNS trauma: a focus on spinal cord injury »
2 – A new figure intitled « Interconnected mechanisms of vasogenic and cytotoxic processes of edema formation » has been added. Minor changes have also been implemented from line 221 to line 227 in order to improve the link between the text and the new schematic.
3 - Several errors in the manuscript have also been corrected
